# Steroid hormone induction of temporal gene expression in *Drosophila* brain neuroblasts generates neuronal and glial diversity

Mubarak Hussain Syed, Brandon Mark, Chris Q Doe*

Institute of Neuroscience, Institute of Molecular Biology, Howard Hughes Medical Institute, University of Oregon, Eugene, United States

**Abstract** An important question in neuroscience is how stem cells generate neuronal diversity. During Drosophila embryonic development, neural stem cells (neuroblasts) sequentially express transcription factors that generate neuronal diversity; regulation of the embryonic temporal transcription factor cascade is lineage-intrinsic. In contrast, larval neuroblasts generate longer ~50 division lineages, and currently only one mid-larval molecular transition is known: Chinmo/Imp/Lin-28+ neuroblasts transition to Syncrip+ neuroblasts. Here we show that the hormone ecdysone is required to down-regulate Chinmo/Imp and activate Syncrip, plus two late neuroblast factors, Broad and E93. We show that Seven-up triggers Chinmo/Imp to Syncrip/Broad/E93 transition by inducing expression of the Ecdysone receptor in mid-larval neuroblasts, rendering them competent to respond to the systemic hormone ecdysone. Importantly, late temporal gene expression is essential for proper neuronal and glial cell type specification. This is the first example of hormonal regulation of temporal factor expression in Drosophila embryonic or larval neural progenitors.

DOI: https://doi.org/10.7554/eLife.26287.001

**\*For correspondence:**
cdoe@uoregon.edu

**Competing interests:** The authors declare that no competing interests exist.

## Introduction

From Drosophila to humans, the brain contains a vast array of morphologically and functionally distinct neurons and glia that arise from a much smaller pool of neural progenitors. How neural stem cells generate neural diversity is a fundamental question that is relevant to many areas of biology. For example, understanding normal neurodevelopmental programs may help design reprogramming protocols to replace specific neurons in clinical trials; may help elucidate principles of connectivity based on shared developmental features; and may help reveal how proliferative neural progenitors avoid tumor formation without differentiating. Drosophila has been a pioneering model system for the study of neural progenitor specification by spatial cues (*Skeath and Thor, 2003*), neural progenitor self-renewal versus differentiation (*Doe, 2008*), stem cell derived tumor formation (*Caussinus and Hirth, 2007*; *Homem et al., 2015*; *Jiang and Reichert, 2014*; *Maurange and Gould, 2005*), and more recently the identification of temporal factors that are sequentially expressed during neural progenitor lineages to increase neural diversity (reviewed in *Kohwi and Doe, 2013*; *Maurange and Gould, 2005*; *Rossi et al., 2017*). In most of the examples cited above, Drosophila studies have revealed conserved mechanisms and/or molecules used in mammals. Here we use Drosophila larval neural progenitors (neuroblasts) to investigate temporal patterning mechanisms that generate neuronal diversity.

There are three types of neuroblasts based on division mode: type 0 neuroblasts make a single neuron with each division; type I neuroblasts make a ganglion mother cell (GMC) with each division, and the GMC typically makes two neurons; and type II neuroblasts make an intermediate neural

progenitor (INP) with each division, and each INP undergoes a short lineage to produce ~6 GMCs and thus ~12 neurons (*Bello et al., 2008*; *Boone and Doe, 2008*; *Bowman et al., 2008*; *Gunnar et al., 2016*; *Wang et al., 2014*). Embryonic type I neuroblasts have short lineages averaging five neuroblast divisions, and they sequentially express five temporal transcription factors: Hunchback, Krüppel, Pdm (the co-expressed Nubbin and Pdm2 proteins), Castor (Cas), and Grainy head (reviewed in *Kohwi and Doe, 2013*; *Maurange and Gould, 2005*; *Rossi et al., 2017*). In addition, the orphan nuclear hormone receptor Seven-up (Svp) acts as a 'switching factor' required for a timely Hunchback-Krüppel transition (*Kanai et al., 2005*; *Mettler et al., 2006*). Most of these temporal transcription factors are necessary and sufficient to specify the identity of neurons born during their neuroblast expression window, although the Cas window can be subdivided further by expression of sub-temporal factors (*Stratmann et al., 2016*). Similarly, larval optic lobe neuroblasts sequentially express six temporal transcription factors that are necessary to generate neuronal diversity in the adult visual system (*Bertet et al., 2014*; *Li et al., 2013*; *Suzuki et al., 2013*). Lastly, INPs sequentially express three temporal transcription factors, two of which are known to specify neural identity (*Bayraktar and Doe, 2013*). Thus, neuroblasts and INPs with short lineages have well-characterized temporal transcription factor cascades that change approximately every cell division, and act to increase neuronal diversity.

In contrast, the central brain type I or II neuroblasts undergo longer lineages of ~50 divisions to generate hundreds of neural progeny (*Ito et al., 2013*; *Yu et al., 2013*). Clonal analysis reveals different levels of morphological diversity among neurons within a single neuroblast lineage (*Ito et al., 2013*; *Yu et al., 2013*) with some neuroblasts making only four cell types (mushroom body neuroblasts) and other neuroblasts making over 40 different cell types (AD or ALad1 neuroblast) (*Kao et al., 2012*). Recent work has provided evidence that central brain neuroblasts change their gene expression profile over time. Our lab and others have shown that larvae express Cas and Svp in type I or type II neuroblasts prior to 48 hr (timing is relative to larval hatching at 0 hr) (*Bayraktar and Doe, 2013*; *Chai et al., 2013*; *Homem et al., 2014*; *Maurange et al., 2008*). In addition, the transcription factor Chinmo and RNA-binding proteins Imp and Lin-28 are detected in young type I neuroblasts prior to 60 hr (*Liu et al., 2015*; *Narbonne-Reveau et al., 2016*). Moving even later, the RNA-binding protein Syncrip and transcription factor Broad have been reported to be expressed in old type I neuroblasts or neurons, respectively from ~60–120 hr (*Liu et al., 2015*; *Maurange et al., 2008*; *Zhou et al., 2009*). Lastly, the steroid hormone ecdysone and the secreted protein Hedgehog are required to terminate neuroblast proliferation after pupariation (*Chai et al., 2013*; *Homem et al., 2014*). Regulation of these temporal gene expression transitions has remained mostly mysterious, but the early factor Svp is required to induce down-regulation of the early Chinmo/Imp/Lin-28 factors at mid-larval stages (*Narbonne-Reveau et al., 2016*).

Despite the recent progress, many important questions remain. First, do larval neuroblasts express additional factors that may be used to generate neuronal diversity? Identifying additional candidate temporal transcription factors would be a major step forward in understanding how neuronal diversity in the adult brain is generated. Second, the role of Svp in regulating larval neuroblast gene expression transitions is poorly understood. Does Svp down-regulate early factors only, or does it activate late factor expression as well, and what are its effector genes? Determining how Chinmo/Imp/Lin-28 are down-regulated is likely to provide insight into how neuronal diversity is generated.

Here we answer each of these questions. We identify candidate temporal transcription factors expressed in neuroblasts that increase the number of molecularly distinct neuroblast temporal profiles; we show that the steroid hormone ecdysone, made outside the CNS, is required for the down-regulation of Chinmo/Imp and activation of Syncrip/Broad/E93 in mid-larval neuroblasts; we show that Svp activates expression of the Ecdysone receptor isoform B1 at mid-larval stages, rendering the neuroblasts competent to respond to the hormone ecdysone; and we show that EcR is required for proper neuronal and glial cell fate specification. Our results are the first example of hormonal regulation of temporal gene expression in neural progenitors, and the first to show that the conserved switching factor Svp can induce neural progenitor competence to respond to an extrinsic hormonal cue.

## Results

### Larval brain neuroblasts undergo an early Chinmo/Imp/Lin-28 to late Broad/Syncrip/E93 transition in gene expression

To determine if larval neuroblasts change their gene expression profile over time, we focused initially on the eight individually identifiable type II neuroblasts (subsequently analyzing all central brain neuroblasts, see below). We took two approaches: we assayed genes known to be expressed in early larval or late larval stages for temporal expression in neuroblasts, and we performed an unbiased transcriptional profiling using the TU-tagging method (*Miller et al., 2009*). The TU-tagging method confirmed our findings on the known genes, and identified additional temporally-regulated genes not previously known to be expressed in neuroblasts (*Figure 1—figure supplement 1*). Here we focus on expression of Cas, Svp, Chinmo, Imp, Lin-28, and Syncrip, which are all known to be temporally expressed in larval neuroblasts (*Bayraktar and Doe, 2013*; *Liu et al., 2015*; *Maurange et al., 2008*). In addition we show that the Ecdysone receptor (EcR), Broad and E93 (Flybase: Eip93F) are also temporally expressed in late larval type II neuroblasts.

The Cas and Svp temporal factors are restricted to the earliest stages of larval type II neuroblast lineages (*Figure 1A–E*). Interestingly, we detected Svp in variable subsets of type II neuroblasts in each brain lobe, consistent with transient, asynchronous expression in all type II neuroblasts. We confirmed that Svp was transiently expressed in all type II neuroblasts by visualizing the more stable *svp-lacZ* reporter in nearly all type II neuroblasts (*Figure 1E*). Three other early factors (Chinmo, Imp, and Lin-28) are expressed in all type II neuroblasts from larval hatching to ~48 hr, becoming undetectable by 72 hr (*Figure 1F–K*). Conversely, the Broad/Syncrip/E93 factors are detected in older type II neuroblasts (*Figure 1L–Q*). There are four known isoforms of Broad (*Zhou et al., 2009*), and we found that Broad-Z1 but not Broad-Z3 was expressed in type II neuroblasts (*Figure 1—figure supplement 2*). Each late factor showed slightly different kinetics of expression: Syncrip was detectable earliest, in all type II neuroblasts by 60 hr, co-expressed with Imp at this stage. Broad was detected in most type II neuroblasts at 60 hr and staying at high levels before declining at 120 hr, and E93 showed gradually increasing expression beginning at 72 hr and remaining at high levels at 120 hr. Thus, these temporal factors can generate seven different molecular profiles during type II neuroblast lineages (summarized in *Figure 1R*); whether all of these molecular differences are functionally important remains to be determined (see Discussion). We conclude that type II neuroblasts change molecular profile over the course of their 120 hr long larval cell lineage, with a striking early-to-late transition from Chinmo/Imp/Lin-28 to Broad/Syncrip/E93 at ~60 hr, midway through their lineage.

### The steroid hormone ecdysone is required for the early-to-late gene expression transition in larval brain neuroblasts

Here we test whether the steroid hormone ecdysone, known to regulate many larval gene expression transitions in multiple tissues (*Faunes and Larraín, 2016*), plays a role in the Chinmo/Imp to Broad/Syncrip/E93 neuroblast gene expression transition. We used three different experiments to test the role of ecdysone: global reduction in ecdysone levels using the *ecdysoneless*[1] (subsequently *ecd^ts*) temperature-sensitive mutation (*Figure 2*); in vitro brain explant culture with or without exogenous ecdysone (*Figure 3*); and type II neuroblast-specific expression of a dominant-negative Ecdysone receptor (*Figure 4*). To reduce global levels of ecdysone we raised *ecd^ts* homozygous larvae at the 29°C restrictive temperature (all larval ages adjusted to match normal 25°C staging, see Materials and methods) and for controls we either assayed the same *ecd^ts* homozygous larvae at the permissive temperature of 18°C or age-matched wild type larvae (controls). As expected, control larvae at 72 hr or 96 hr had type II neuroblasts that expressed the late factors Syncrip, Broad and E93, but not the early factors Chinmo and Imp (*Figure 2A–E*; quantified in K). In contrast, *ecd^ts* homozygous larvae at 29°C had type II neuroblasts that persistently expressed the early factors Chinmo/Imp and lacked the late factors Syncrip/Broad/E93 (*Figure 2F–J*; quantified in K). We conclude that systemic reduction of ecdysone levels blocks the Chinmo/Imp to Broad/Syncrip/E93 gene expression transition in type II neuroblasts.

Loss of ecdysone signaling could block cell cycle progression of type II neuroblasts, which could prevent the early-to-late gene expression transition. To test this hypothesis, we isolated larval brains

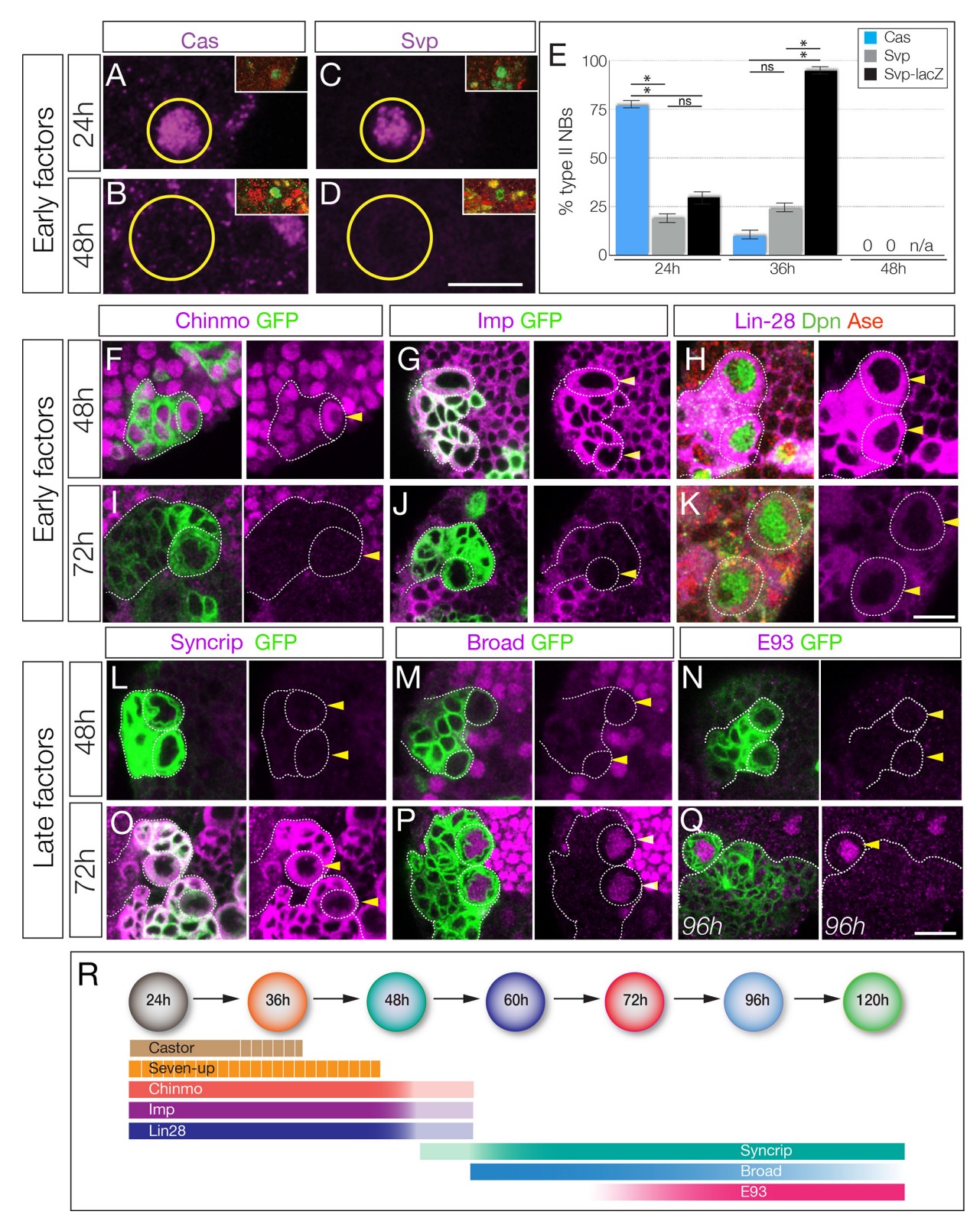

**Figure 1.** Identification of temporally expressed proteins in larval type II neuroblasts. (**A–E**) Cas and Svp are expressed from 24–36 hr (**A,C,E**) but not at 48 hr (**B,D,E**). Neuroblasts, outlined. (**F–K**) Early factors. Chinmo, Imp, and Lin-28:GFP (Lin-28) are detected in neuroblasts at 48 hr but not at 72 hr. (**L–Q**) Late factors. Syncrip, Broad, and E93 are not detected in neuroblasts at 48 hr but are present at 72 hr or 96 hr. (**R**) Summary of temporal factor expression. Dashed bars indicate asynchronous expression during the indicated temporal window. Gradients indicate graded change in expression

*Figure 1 continued on next page*

*Figure 1 continued*

levels. In all panels, temporal factors are in magenta, and type II neuroblasts are identified by *wor-gal4 ase-gal80 UAS-mcd8:GFP* transgene expression (GFP, green, outlined) or as Dpn+ Ase- (green/red, respectively). Arrowhead, neuroblasts. For each panel n > 10 neuroblasts scored. Scale bar, 10 μm.

DOI: https://doi.org/10.7554/eLife.26287.002

The following figure supplements are available for figure 1:

**Figure supplement 1.** TU-tagging to identify temporally expressed genes in type II neuroblasts and their progeny.

DOI: https://doi.org/10.7554/eLife.26287.003

**Figure supplement 2.** Broad-Z1 but not Broad-Z3 is expressed in type II neuroblasts.

DOI: https://doi.org/10.7554/eLife.26287.004

and cultured them in vitro from 48–72 hr (across the early-to-late transition) with or without the bio-active form of ecdysone (20-hydroxy-ecdysone; 20HE). We used live imaging to measure cell cycle times, as well as assayed for expression of representative early or late factors. Larval brains cultured with added ecdysone are similar to wild type in down-regulating the early factor Chinmo and expressing the late factor Broad by the end of the culture period at 72 hr (*Figure 3A–B*; quantified in H; *Video 1*). In contrast, larval brains cultured without ecdysone showed persistent Chinmo expression and failed to express Broad (*Figure 3C–D*; quantified in H; *Video 2*). Importantly, the cell cycle times of type II neuroblasts are indistinguishable with or without ecdysone (*Figure 3E–F*, quantified in G; *Videos 1* and *2*), and similar to published type II neuroblast cell cycle times (*Homem et al., 2013*). Taking these results together with our in vivo ecdysone experiments, we

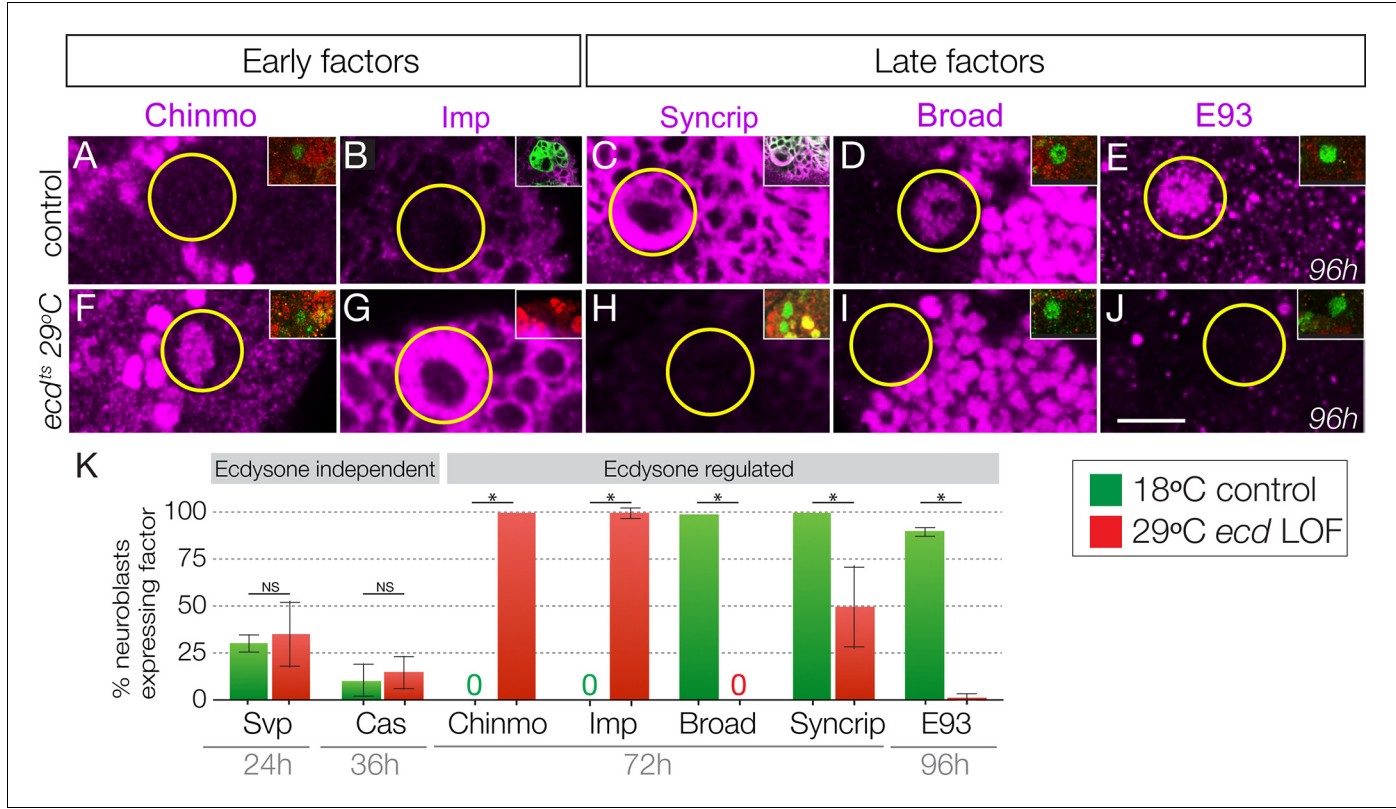

**Figure 2.** Ecdysone hormone is required for the early-to-late temporal factor transition. (A–E) Control *ecd^ts*/*deficiency* larvae at 18°C show normal temporal factor expression in type II neuroblasts (circled): the early factors Chinmo and Imp are off at 72 hr (A–B) and the late factors Syncrip, Broad, and E93 are on at 72 hr and 96 hr (C–E). (F–J) Loss of ecdysone in *ecd^ts* /*deficiency* larvae at 29°C shows failure to down-regulate the early factors Chinmo and Imp (F–G) and failure to activate the late factors Syncrip, Broad and E93 (H–J) in type II neuroblasts (circled). (K) Quantification. n > 10 for each bar. Asterisk, p<0.003. In all panels, times are adjusted to the equivalent larval stage at 25°C, type II neuroblasts are identified as Dpn+ Ase- or large cells expressing *wor-gal4 ase-gal80 UAS-mcd8:GFP* (green in insets). Scale bar, 10 μm.

DOI: https://doi.org/10.7554/eLife.26287.005

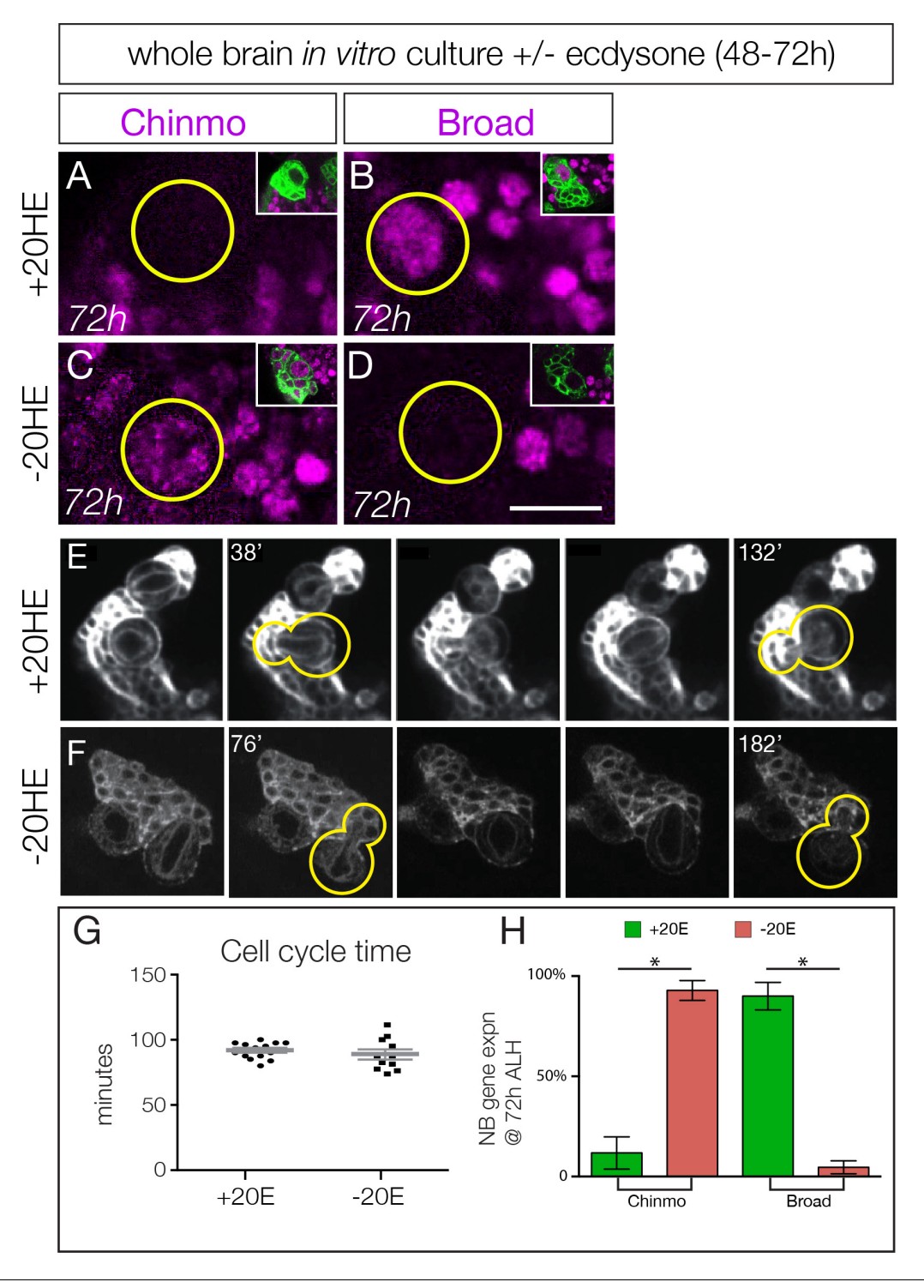

**Figure 3.** Ecdysone hormone activates neuroblast expression of Chinmo and Broad in isolated brain cultures. (A–B) Isolated larval brains cultured with added 20-hydroxy-ecdysone (+20 HE) from 48–72 hr show normal down-regulation of the early factor Chinmo (A) and activation of the late factor Broad (B). (C–D) Isolated larval brains cultured without added 20-hydroxy-ecdysone (−20HE) from 48–72 hr fail to down-regulate Chinmo (C) or activate Broad (D). (E–F) Live imaging of isolated larval brains from 48–72 hr cultured with ecdysone (+20 HE) or without ecdysone (−20HE). In each case two neuroblasts and their progeny were imaged; successive telophase stages are shown for one neuroblast (yellow outline); note that for both +20 HE and −20HE the cell cycle time is ~100 min (see timestamps). See *Videos 1 and 2*. (G) Quantification of the experiment shown in E–F. (H) Quantification of

*Figure 3 continued on next page*

*Figure 3 continued*
the experiment shown in **A–D**. In all panels, type II neuroblasts are identified as large cells expressing *wor-gal4 ase-gal80 UAS- mcd8:GFP* (green in insets, **A–F**; white in **G–H**), n $\geq$ 10 per experiment. Asterisk, p<0.003. Scale bar, 10 µm.
DOI: https://doi.org/10.7554/eLife.26287.006

conclude that the steroid hormone ecdysone induces the early-to-late Chinmo/Imp to Broad/Syncrip/E93 gene expression transition in type II neuroblasts.

## The ecdysone receptor EcR-B1 is expressed concurrent with, and required for, the early-to-late gene expression transition in larval brain neuroblasts

The ability of ecdysone signaling to trigger a major gene expression transition at ~60 hr could be due to a peak of ecdysone signaling at that time, or the lack of a signaling pathway component prior to that time. Ecdysone is present at all larval stages (*Kozlova and Thummel, 2000*), suggesting the latter mechanism. Ecdysone signaling is quite direct, requiring the Ecdysone receptor (EcR-A, -B1, or -B2 isoforms) and the ubiquitous co-receptor Ultraspiracle (*Lee et al., 2000*). We found that EcR-B1 was strongly detected in the nuclei of type II neuroblasts from 56 hr to at least 120 hr; prior to 56 hr neuroblasts did not express any EcR isoform (*Figure 4B*; *Figure 4—figure supplement 1*). To determine if EcR-B1 expression was induced by ecdysone signaling (e.g. via a different EcR receptor isoform) we assayed *ecd^{ts}* mutants at non-permissive temperature and assayed for EcR-B1 expression; we also assayed for EcR-B1 expression in brain explants cultured with or without exogenous ecdysone. In both experiments, we found that expression of EcR-B1 was not dependent on ecdysone signaling (*Figure 4C–F*). Note that normal EcR expression during the culture window provides evidence that there is no general developmental delay in brains lacking ecdysone signaling, despite failure to undergo the Chinmo/Imp to Broad/Syncrip/E93 transition. We conclude that an ecdysone-independent pathway activates EcR-B1 expression at 56 hr, the time of the Chinmo/Imp to Broad/Syncrip/E93 gene expression transition (summarized in *Figure 4M*).

To determine if EcR was required for the Chinmo/Imp to Broad/Syncrip/E93 gene expression transition, we used *wor-gal4 ase-gal80* to drive expression of a dominant-negative Ecdysone receptor (*EcR-B1^{W650A}*; subsequently *EcR^{DN}*) specifically in type II neuroblasts, and assayed representative early and late factors. We found that neuroblasts expressing *EcR^{DN}* showed completely penetrant persistent expression of the early factors Chinmo and Imp, and reduced or no expression of the late factors Syncrip (reduced) and E93 (absent) (*Figure 4G–I,K*; quantified 4L). Surprisingly, the late factor Broad was normally expressed (*Figure 4H*; quantified in 4L; see Discussion), again suggesting no general developmental delay. We conclude that ecdysone signaling acts via EcR-B1 within type II neuroblasts to promote the Chinmo/Imp/Lin-28 to Broad/Syncrip/E93 gene expression transition.

## The Seven-up nuclear hormone receptor activates EcR in larval brain neuroblasts

Previous work has shown that *svp* mutant clones fail to down-regulate Chinmo/Imp in late-born neurons (*Narbonne-Reveau et al., 2016*). Svp could promote EcR expression, or act in parallel to EcR to down-regulate Chinmo/Imp. To distinguish between these alternatives, we examined *svp* mutant clones for EcR expression. We induced *svp* mutant clones at 0–4 hr and assayed them at 96 hr. As expected, we observed a highly penetrant failure to down-regulate expression of the early factors Chinmo/Imp (*Figure 5A–B*; quantified in G). More importantly, we found a complete loss of EcR-B1 and the late factors Broad/Syncrip/E93 (*Figure 5C–F*; quantified in G). We conclude that Svp is required to induce EcR expression, and that EcR expression renders neuroblasts competent to respond to ecdysone signaling.

Svp could directly activate EcR expression, or alternatively Svp could terminate Cas expression and Cas could repress EcR (double repression motif). We assayed *svp* mutant clones for Cas expression, and did not observe prolonged Cas expression (*Figure 5—figure supplement 1*). In addition, *cas* mutant clones showed no change in early or late temporal factor expression (data not shown; T.

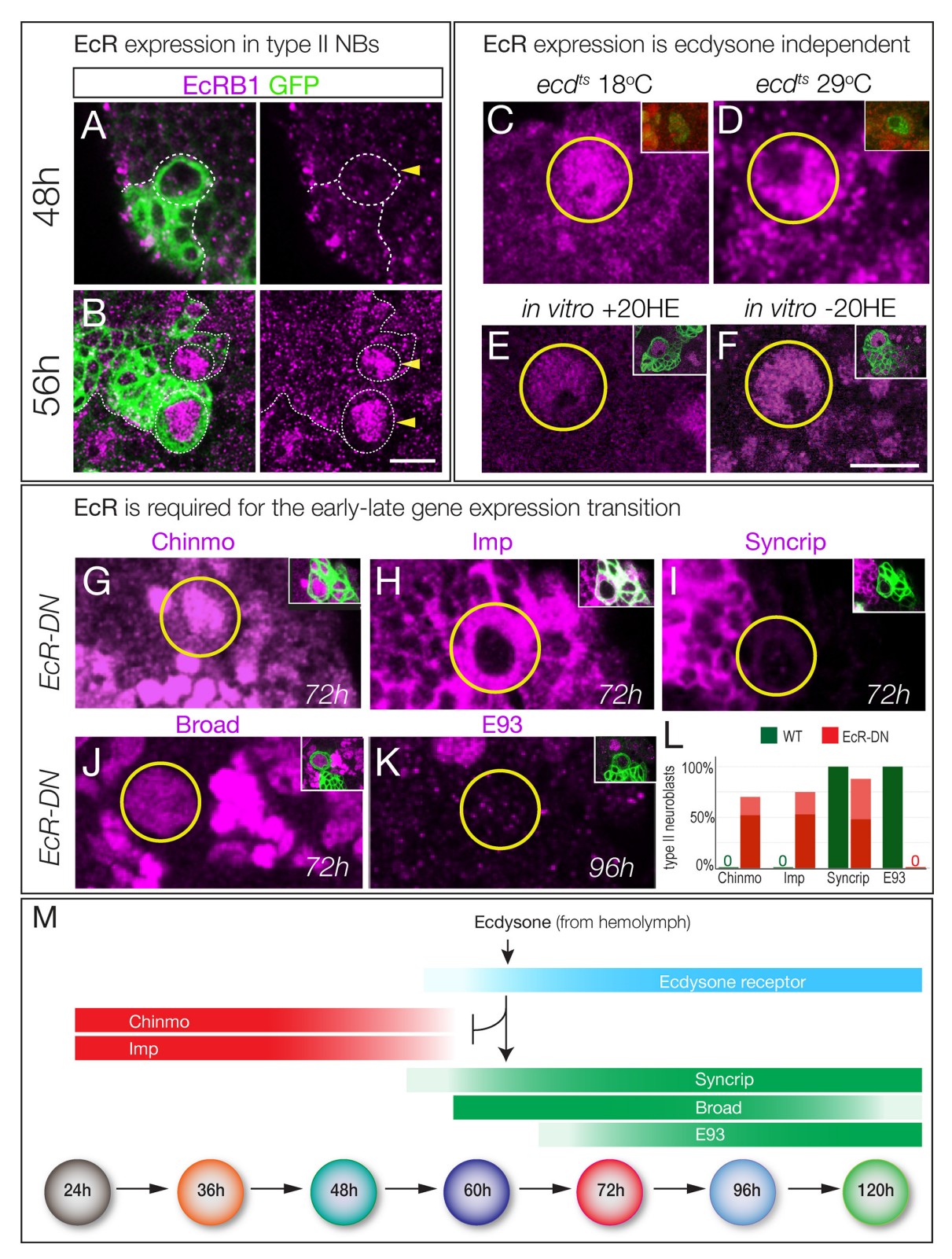

**Figure 4.** Ecdysone receptor expression and function. (**A–B**) EcR-B1 is first detected at ~56 hr in most type II neuroblasts. (**C–F**) EcR-B1 expression is ecdysone-independent. (**C–D**) EcR-B1 is activated normally in *ecd^ts* mutants at both permissive (18°C) and restrictive (29°C) temperatures by 72 hr. (**E–F**) EcR-B1 expression is activated normally in isolated brains cultured from 48–72 hr with (**E**) or without (**F**) added 20-hydroxy-ecdysone (20HE). (**G–K**) Expression of an EcR dominant negative transgene in type II neuroblasts (*wor-gal4 ase-gal80 UAS-mcd8:GFP UAS-EcR^DN*) results in persistent

*Figure 4 continued on next page*

*Figure 4 continued*

expression of the early factors Chinmo and Imp (**G,H**) and failure to express the late factors Syncrip and E93 (**I,K**). Surprisingly, the late factor Broad is still expressed (**J**). (**L**) Quantification. Percent of type II neuroblasts expressing the indicated factors at the indicated levels (dark red, strong expression; light red, weak expression; dark green, strong expression). All data are from 72 hr larvae except E93 is from 96 hr larvae. (**M**) Summary: ecdysone signaling via EcR-B1 terminates expression of early factors and activates expression of late factors. In all panels, type II neuroblasts are identified by expression of *wor-gal4 ase-gal80 UAS-mcd8:GFP* (left panels or insets), developmental times are adjusted to the equivalent time at 25°C. n > 10 for each experiment. Scale bar 10 μm.

DOI: https://doi.org/10.7554/eLife.26287.007

The following figure supplement is available for figure 4:

**Figure supplement 1.** Ecdysone receptor isoform expression in type II neuroblasts.

DOI: https://doi.org/10.7554/eLife.26287.008

Lee, personal communication). We conclude that Svp uses a Cas-independent mechanism to activate EcR expression in type II neuroblasts.

## Syncrip represses Imp expression to allow down-regulation of Chinmo in larval neuroblasts

We wanted to characterize the genetic interactions occurring downstream of the Ecdysone receptor, to better understand the mechanism of the early-to-late gene expression transition. Does each affected gene respond independently to EcR, or is there a cascade of interactions occurring downstream of a single primary EcR target gene?

First, we tested whether the Imp and Syncrip proteins are downstream of EcR-B1. We observed normal EcR-B1 timing and levels in both *Imp* and *Syncrip* mutants (***Figure 6A–C***), confirming that they act in parallel or downstream of EcR-B1. Second, we tested whether Imp and Syncrip cross-repress each other, as has been shown for mushroom body neuroblasts (***Liu et al., 2015***). We found that *Syncrip* mutants had prolonged Imp expression in type II neuroblasts, but that *Imp* mutants did not show precocious Syncrip expression in the six DM1-DM6 type II neuroblasts (***Figure 6D–E***); there was variable precocious expression in one of the DL1/2 type II neuroblasts at 48 hr (data not shown). We conclude that the regulatory interactions between Imp and Syncrip are neuroblast-specific.

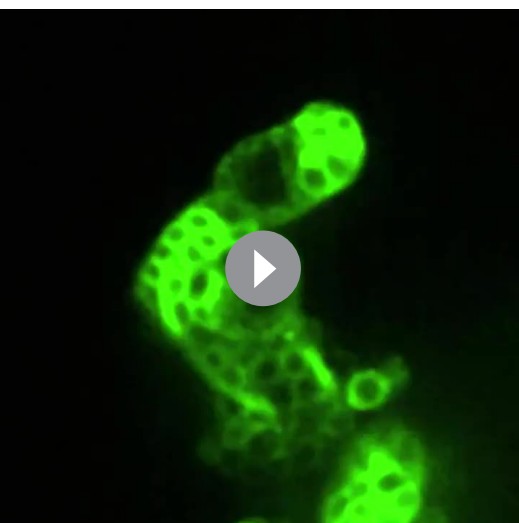 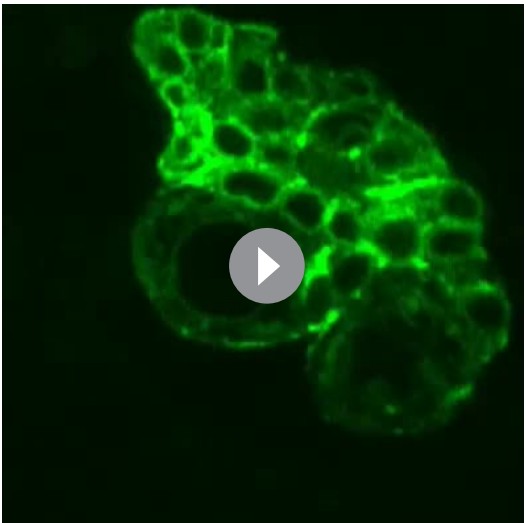

**Video 1.** Explanted larval brain cultured in vitro from 48–72 hr with added 20-hydroxy-ecdysone. Two type II neuroblasts are shown expressing *wor-gal4 ase-gal80 UAS-myr:GFP* (see Materials and methods). Still panels from video shown in ***Figure 3E***, and cell cycle times quantified in ***Figure 3G***.

DOI: https://doi.org/10.7554/eLife.26287.009

**Video 2.** Explanted larval brain cultured in vitro from 48–72 hr without 20-hydroxy-ecdysone. Two type II neuroblasts are shown expressing *wor-gal4 ase-gal80 UAS-myr:GFP* (see Materials and methods). Still panels from video shown in ***Figure 3F***, and cell cycle times quantified in ***Figure 3G***.

DOI: https://doi.org/10.7554/eLife.26287.010

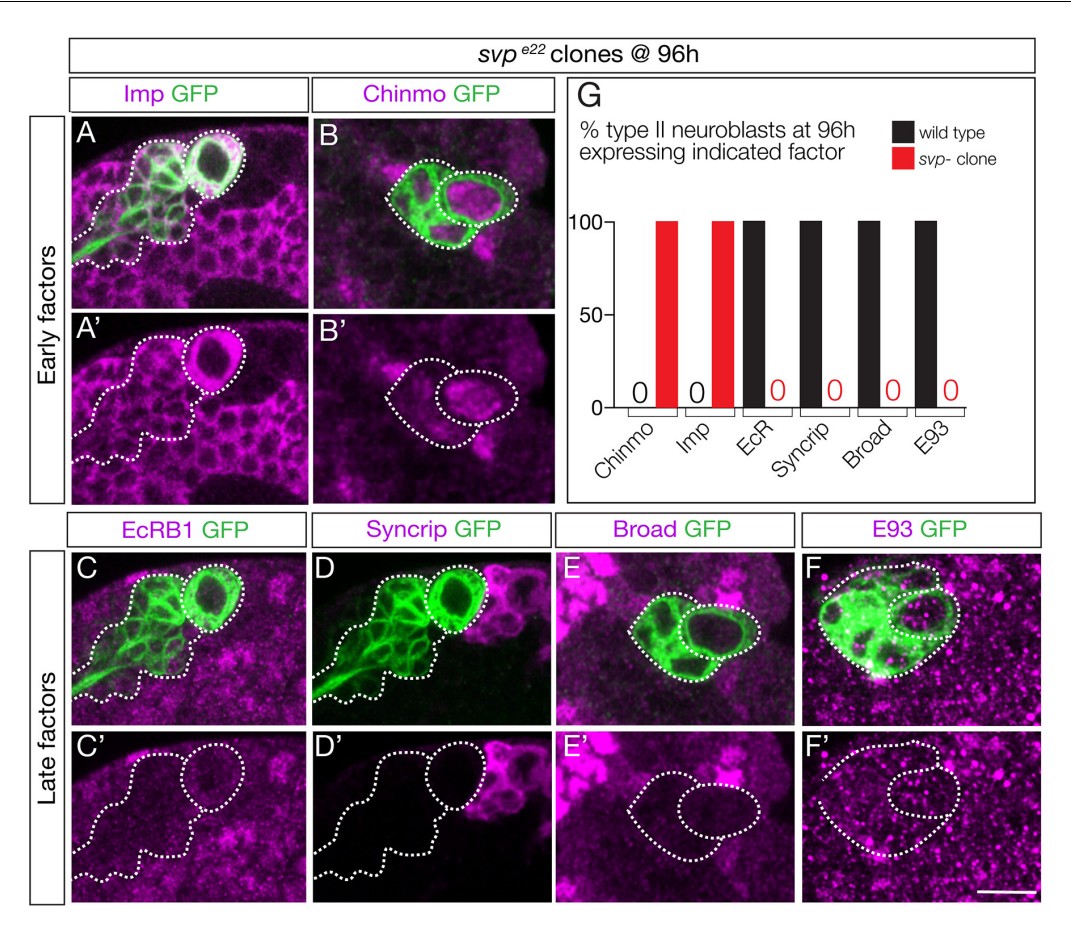

**Figure 5.** Seven-up activates expression of the Ecdysone receptor in type II neuroblasts. (**A–F**) *svp* mutant MARCM clones (GFP+, green and outlined) induced at 0–4 hr and assayed at 96 hr for the indicated factors. (**G**) Quantification (red, *svp* mutant clone; black, wild type *UAS-FLP actin-FRT-stop-FRT-gal4; wor-gal4 ase-gal80; UAS-mCD8: GFP*). 100% of mutant type II neuroblasts fail down-regulate the early factors Imp and Chinmo, and fail to activate the late factors Syncrip, EcR-B1, E93 and Broad. Number of *svp* mutant clones scored: Imp n = 11, Chinmo n = 4, Syncrip n = 19, EcR-B1 n = 11, E93 n = 2, Broad n = 4. Number of wild type neuroblasts scored n > 10 for each marker. In all panels, type II neuroblasts are identified by expression of *wor-gal4 ase-gal80 UAS-mcd8:GFP* (outlined). Scale bar 10 μm.
DOI: https://doi.org/10.7554/eLife.26287.011

The following figure supplement is available for figure 5:

**Figure supplement 1.** Cas is not required to activate Svp expression, and Svp is not required to terminate Cas expression.
DOI: https://doi.org/10.7554/eLife.26287.012

Third, we assayed *Imp* and *Syncrip* mutants for changes in early and late factor expression. Whereas *Imp* mutants had no change in expression of the early factor Chinmo or late factors Broad and E93 (**Figure 6F–H**), *Syncrip* mutants showed prolonged expression of early factor Chinmo but normal expression of the late factors Broad and E93 (**Figure 6I–K**), leading to a novel Chinmo+ Broad+ co-expression molecular profile (**Figure 6N**). Lastly, we tested for cross-repression between the early factor Chinmo and the late factor Broad, as these proteins are typically mutually exclusive in both neuroblasts and their neuronal progeny (**Maurange et al., 2008**; **Narbonne-Reveau et al., 2016**; **Zhu et al., 2006**)(**Figure 6—figure supplement 1**). We found that Broad was expressed normally in *chinmo[1]* mutant clones and Chinmo was expressed normally in *broad[npr3]* mutants (**Figure 6L,M**). Taken together, our results support a model in which EcR independently activates all known late factors, with the late factor Syncrip required to repress early factor expression.

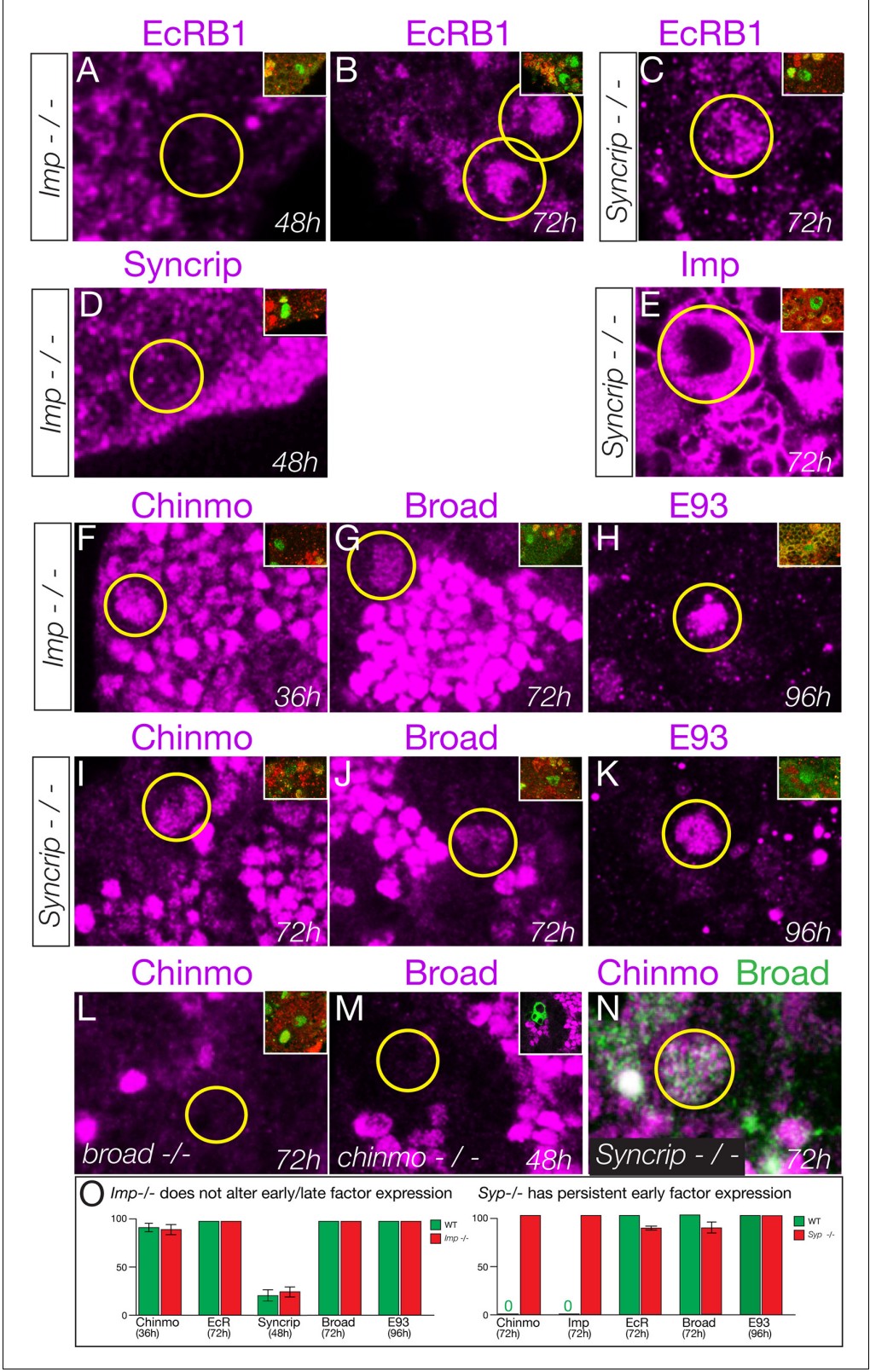

**Figure 6.** Syncrip and Imp function downstream of the Ecdysone receptor in type II neuroblasts. (A–C) *Imp* mutants *(imp^{G0072} / imp^{G0072})* and *Syncrip* mutants *(Syncrip ^{f03775}/ deficiency)* show normal expression of EcR in type II neuroblasts: off at 48 hr and on at 72 hr. (D) *Imp* mutants *(Imp^{G0072} / Imp^{G0072})* do not precociously upregulate Syncrip in type II neuroblasts at 48 hr. (E) *Syncrip* mutants *(Syncrip ^{f03775}/ deficiency)* show prolonged

*Figure 6 continued on next page*

*Figure 6 continued*

expression of Imp in type II neuroblasts at 72 hr. (**F–H**) *Imp* mutants (*Imp$^{G0072}$ / Imp$^{G0072}$*) show normal temporal expression of Chinmo, Broad, or E93 in type II neuroblasts. (**I–K**) *Syncrip* mutants (*Syncrip$^{f03775}$/ deficiency*) show prolonged expression of Chinmo but normal expression of the late factors Broad and E93 in type II neuroblasts. (**L**) *broad* mutants (*broad$^{npr3}$/ broad$^{npr3}$*) have normal Chinmo expression: absent from 72 hr type II neuroblasts. (**M**) *chinmo* mutants (*chinmo$^1$* mutant clones) have normal Broad expression: absent from 48 hr type II neuroblasts. (**N**) *Syncrip* mutants (*Syncrip$^{f03775}$/ deficiency*) have type II neuroblasts that abnormally co-express Chinmo and Broad at 72 hr. (**O**) Quantification. Percent of type II neuroblasts expressing the indicated factors at the indicated timepoints. Insets identify the pictured type II neuroblast (green) based on its expression of Dpn (green) not Ase (red) or by expression of *wor-gal4 ase-gal80 UAS-mcd8:GFP* (green). n > 5 for all panels. Scale bar, 10 µm.
DOI: https://doi.org/10.7554/eLife.26287.013

The following figure supplement is available for figure 6:

**Figure supplement 1.** Chinmo and Broad have mutually exclusive expression in neuroblasts and neurons.
DOI: https://doi.org/10.7554/eLife.26287.014

## Late temporal transcription factors are required to specify adult neuronal identity

The functional analysis of all eight candidate temporal transcription factors is beyond the scope of this study, in part due to the absence of markers for neurons or glia produced during each specific window of gene expression. Nevertheless, two markers label progeny born either early or late in type II neuroblast lineages: the glial marker Repo stains a pool of early-born glia whereas the neuronal marker Brain-specific homeobox (Bsh) stains late-born neurons within type II neuroblast lineages (*Bayraktar and Doe, 2013*). To determine if the late temporal factors play a role in repressing early-born Repo+ glial identity or inducing late-born Bsh+ neuronal identity, we expressed the EcR dominant negative transgene specifically and permanently in type II neuroblast lineages (see 'lineage tracing' Materials and methods). We found that reducing EcR caused an increase in the early marker Repo (*Figure 7A,B*; quantified in C) and a decrease in the late marker Bsh (*Figure 7D,E*; quantified in F), consistent with a role for EcR or a downstream late temporal factor in suppressing early-born glial identity and promoting late-born neuronal identity.

## The ecdysone-dependent Chinmo/Imp to Broad/Syncrip/E93 gene expression transition is widely used by central brain type I neuroblasts

We have focused on a small pool of type II neuroblasts because they are individually identifiable and tools to mark and manipulate them are available. Yet the majority of central brain neuroblasts are type I neuroblasts (~95 per lobe). Here we test whether the early-to-late Chinmo/Imp to Syncrip/Broad/E93 gene expression transition occurs in this populations of neuroblasts, by assaying representative early or late factors in larvae with reduced ecdysone signaling (*ecd$^{ts}$* mutants). In *ecd$^{ts}$* mutants raised at the permissive temperature to allow ecdysone signaling, we observe normal down-regulation of the early factor Chinmo and activation of the late factors Broad and E93 at 96 hr in type I central brain neuroblasts (*Figure 8A–C*; quantified in G). In contrast, *ecd$^{ts}$* mutant larvae placed at restrictive temperature to block ecdysone signaling showed persistent expression of the early factor Chinmo and failure to activate the late factors Broad and E93 at 96 hr (*Figure 8D–F*; quantified in G). Notably, we find that Syncrip is expressed in a small number of type I neuroblasts (~10) prior to widespread EcR expression at 56 hr (*Figure 8—figure supplement 1*). It is likely that these neuroblasts use an EcR-B1-independent mechanism for activating Syncrip expression. We conclude that most central brain neuroblasts undergo an ecdysone-dependent early-to-late Chinmo/Imp to Syncrip/Broad/E93 gene expression transition.

## Discussion

Here we show that the steroid hormone ecdysone is required to trigger a major gene expression transition at mid-larval stages: central brain neuroblasts transition from Chinmo/Imp to Broad/Syncrip/E93. Furthermore, we show that Svp activates expression of EcR-B1 in larval neuroblasts, which gives them competence to respond to ecdysone signaling, thereby triggering this gene expression transition. Although a global reduction of ecdysone levels is likely to have pleiotropic effects on

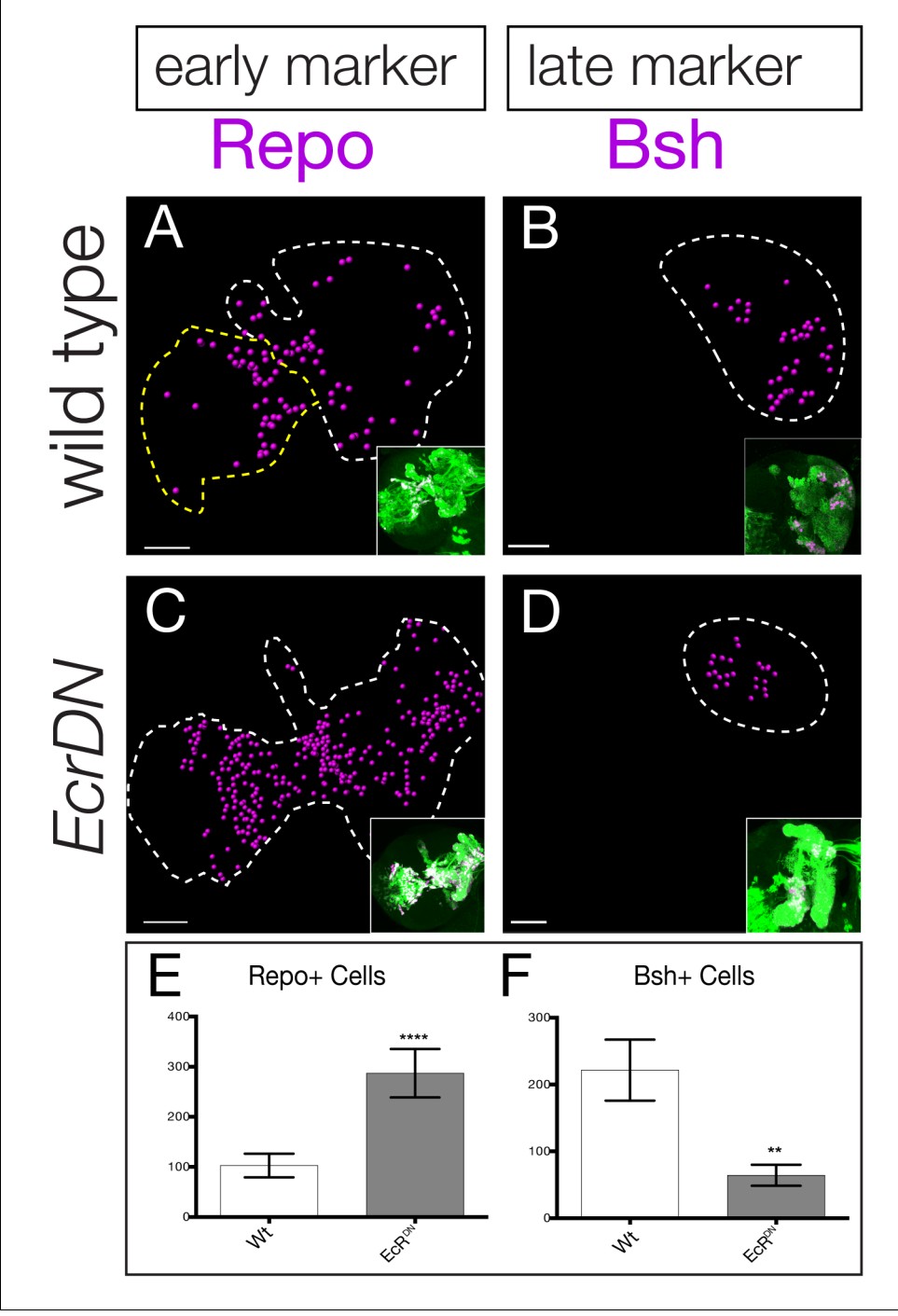

**Figure 7.** Late temporal transcription factors specify neuronal and glial identity. (**A,B**) Wild type or (**C,D**) *EcR*<sup>DN</sup> brains at 0 hr after puparium formation. The inset shows GFP+ cells permanently marking the type II neuroblast lineage (*wor-gal4 ase-gal80 UAS-FLP actin-FRT-stop-FRT-gal4 UAS-mCD8:GFP*; green) which is circled with dashed lines in the main figure. Repo+ or Bsh+ nuclei located within the volume of the type II progeny were identified using Imaris and represented as magenta spheres. This provides the optimal way to visualize cell numbers within the three dimensional GFP+ volume. (**E,F**) Quantification. **p<0.01, ****p<0.0001.
DOI: https://doi.org/10.7554/eLife.26287.015

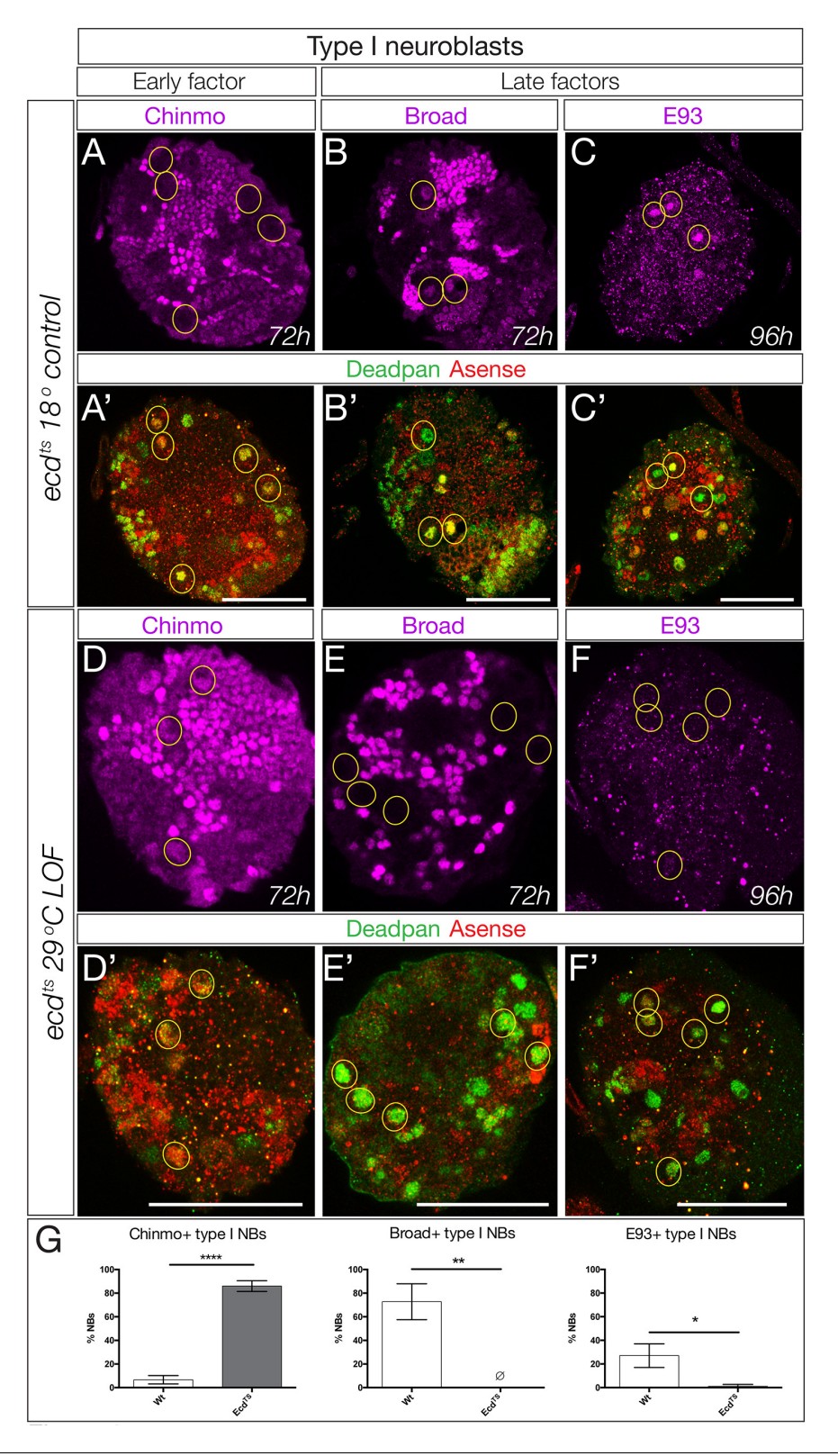

**Figure 8.** Ecdysone is required for early to late temporal factor transition in type I neuroblasts. (**A–C**) Control brains (*ecd-ts/deficiency* at 18°C) at the indicated timepoint. (**A–C**) Normal down-regulation of the early factor Chinmo and activation of the late factors Broad and E93. (**E–G**) Experimental brains with reduced ecdysone (*ecd-ts/deficiency* at 29°C) at the indicated timepoint. (**E–G**) Abnormal prolonged expression of the early factor Chinmo

*Figure 8 continued on next page*

*Figure 8 continued*

and failure to activation of the late factors Broad and E93. (D,H) Quantification. n = 6 brain lobes; *p<0.03, **p<0.03; ****p<0.0001. In all panels, central brain type I neuroblasts are identified as Dpn+Ase+ (subset outlined), and times are adjusted to the equivalent larval stage at 25°C as described in the Materials and methods. Note that *ecd^{ts}/deficiency* brains are smaller than control brains primarily due to severe loss of the optic lobe. Scale bar, 50 μm.

DOI: https://doi.org/10.7554/eLife.26287.016

The following figure supplement is available for figure 8:

**Figure supplement 1.** Syncrip is expressed in a subset of central brain neuroblasts at 36 hr and 48 hr.

DOI: https://doi.org/10.7554/eLife.26287.017

larval development, we have performed multiple experiments to show that the absence or delay in late temporal factor expression following reduced ecdysone signaling is not due to general developmental delay. First, the EcR gene itself is expressed at the normal time (~56 hr) in the whole organism *ecdysoneless[1]* mutant, arguing strongly against a general developmental delay. Second, a type II neuroblast *seven-up* mutant clone shows a complete failure to express EcR and other late factors, in the background of an entirely wild type larvae; this is perhaps the strongest evidence that the phenotypes we describe are not due to a general developmental delay. Third, lineage-specific expression of EcR dominant negative leads to loss of Syncrip and E93 expression without affecting Broad expression; the normal Broad expression argues against a general developmental delay. Fourth, we used live imaging to directly measure cell cycle times and found that lack of ecdysone did not slow neuroblast cell cycle times. Taken together, these data support our conclusion that ecdysone signaling acts directly on larval neuroblasts to promote an early-to-late gene expression transition.

## Ecdysone is the first neuroblast-extrinsic cue known to regulate temporal gene expression

The role of ecdysone in regulating developmental transitions during larval stages has been well studied; it can induce activation or repression of suites of genes in a concentration dependent manner (reviewed in *Thummel, 2001*; *Yamanaka et al., 2013*). Ecdysone induces these changes through a heteromeric complex of EcR and the retinoid X receptor homolog Ultraspiracle (*King-Jones and Thummel, 2005*; *Yamanaka et al., 2013*). Ecdysone is required for termination of neuroblast proliferation at the larval/pupal transition (*Homem et al., 2014*), and is known to play a significant role in remodeling of mushroom body neurons and at neuromuscular junctions (*Awasaki and Lee, 2011*; *Kucherenko and Shcherbata, 2013*; *Lee et al., 2000*; *Schubiger et al., 1998*; *Yu and Schuldiner, 2014*). Here we add to this list another function: to trigger a major gene expression transition in mid-larval brain neuroblasts.

Does ecdysone signaling provide an extrinsic cue that synchronizes larval neuroblast gene expression? We do not see good coordination of late gene expression, arguing against synchronization. For example, Syncrip can be detected in many neuroblasts by 60 hr, whereas Broad appears slightly later at ~72 hr, and E93 is only detected much later at ~96 hr, by which time Broad is low. This staggered expression of ecdysone target genes is reminiscent of early and late ecdysone-inducible genes in other tissues (*Baehrecke, 1996*). In addition, for any particular temporal factor there are always some neuroblasts expressing it prior to others, but not in an obvious pattern. It seems the exact time of expression can vary between neuroblasts. Whether the pattern of response is due to different neuroblast identities, or a stochastic process, remains to be determined.

We have previously shown that the Hunchback-Krüppel-Pdm-Castor temporal gene transitions within embryonic neuroblasts are regulated by neuroblast-intrinsic mechanisms: they can occur normally in neuroblasts isolated in culture, and the last three factors are sequentially expressed in $G_2$-arrested neuroblasts (*Grosskortenhaus et al., 2005*). Similarly, optic lobe neuroblasts are likely to undergo neuroblast-intrinsic temporal transcription factor transitions, based on the observation that these neuroblasts form over many hours of development and undergo their temporal transitions asynchronously (*Bertet et al., 2014*; *Hasegawa et al., 2011*; *Li et al., 2013*; *Suzuki et al., 2013*). In contrast, we show here that ecdysone signaling triggers a mid-larval transition in gene expression in all central brain neuroblasts (both type I and type II). Although ecdysone is present at all larval

stages, it triggers central brain gene expression changes only following Svp-dependent expression of EcR-B1 in neuroblasts. Interestingly, precocious expression of EcR-B1 (*worniu-gal4 UAS-EcR-B1*) did not result in premature activation of the late factor Broad, despite the forced expression of high EcR-B1 levels in young neuroblasts (data not shown). Perhaps there is another required factor that is also temporally expressed at 56 hr. We also note that reduced ecdysone signaling in *ecd*[ts] mutants or following EcR[DN] expression does not permanently block the Chinmo/Imp to Broad/Syncrip/E93 transition; it occurs with variable expressivity at 120–160 hr animals (pupariation is significantly delayed in these *ecd*[ts] mutants), either due to a failure to completely eliminate ecdysone signaling or the presence of an ecdysone-independent mechanism.

We find a small but reproducible difference in the effect of reducing ecdysone levels using the biosynthetic pathway mutant *ecd*[ts] versus expressing a dominant negative EcR in type II neuroblasts. The former genotype shows a highly penetrant failure to activate Broad in old neuroblasts, whereas the latter genotype has normal expression of Broad (despite failure to down-regulate Chinmo/Imp or activate E93). This may be due to failure of the dominant negative protein to properly repress the *Broad* gene. Differences between EcR[DN] and other methods of reducing ecdysone signaling have been noted before (*Brown et al., 2006*; *Homem et al., 2014*).

## Seven-up, but not Castor, is required to activate EcR receptor expression

Drosophila Svp is an orphan nuclear hormone receptor (*Mlodzik et al., 1990*) with an evolutionarily conserved role in promoting a switch between temporal identity factors (*Kohwi and Doe, 2013*). In Drosophila, Svp it is required to switch off *hunchback* expression in embryonic neuroblasts (*Benito-Sipos et al., 2011*; *Kanai et al., 2005*; *Mettler et al., 2006*), and in mammals the related COUP-TF1/2 factors are required to terminate early-born cortical neuron production (*Naka et al., 2008*), as well as for the neurogenic to gliogenic switch (*Barnabé-Heider et al., 2005*; reviewed in *Kohwi and Doe, 2013*). Here we show that Svp is required for activating expression of EcR, which drives the mid-larval switch in gene expression from Chinmo/Imp to Syncrip/Broad/E93 in central brain neuroblasts. Our results are supported by independent findings that svp mutant clones lack expression of Syncrip and Broad in old type II neuroblasts (T. Lee, personal communication). Interestingly, Svp is required for neuroblast cell cycle exit at pupal stages (*Maurange et al., 2008*), but how the early larval expression of Svp leads to pupal cell cycle exit was a mystery. Our results provide a satisfying link between these findings: we show that Svp activates expression of EcR-B1, which is required for the expression of multiple late temporal factors in larval neuroblasts. Any one of these factors could terminate neuroblast proliferation at pupal stages, thereby explaining how an early larval factor (Svp) can induce cell cycle exit five days later in pupae. It is interesting that one orphan nuclear hormone receptor (Svp) activates expression of a second nuclear hormone receptor (EcR) in neuroblasts. This motif of nuclear hormone receptors regulating each other is widely used in Drosophila, C. elegans, and vertebrates (*Boulanger et al., 2011*; *Faunes and Larraín, 2016*; *Gissendanner et al., 2004*; *Lam et al., 1999*; *Thummel, 2001*; *Yamanaka et al., 2013*; *Zelhof et al., 1995*).

The position of the Svp+ neuroblasts varied among the type II neuroblast population from brain-to-brain, suggesting that Svp may be expressed in all type II neuroblasts but in a transient, asynchronous manner. This conclusion is supported by two findings: the *svp-lacZ* transgene, which encodes a long-lived β-galactosidase protein, can be detected in nearly all type II neuroblasts; and our finding that Svp is required for EcR expression in all type II neuroblasts, consistent with transient Svp expression in all type II neuroblasts. It is unknown what activates Svp in type II neuroblasts; its asynchronous expression is more consistent with a neuroblast-intrinsic cue, perhaps linked to the time of quiescent neuroblast re-activation, than with a lineage-extrinsic cue. It would be interesting to test whether Svp expression in type II neuroblasts can occur normally in isolated neuroblasts cultured in vitro, similar to the embryonic temporal transcription factor cascade (*Grosskortenhaus et al., 2005*).

Castor and its vertebrate homolog Cas-Z1 specify temporal identity in Drosophila embryonic neuroblast lineages and vertebrate retinal progenitor lineages, respectively (*Grosskortenhaus et al., 2006*; *Mattar et al., 2015*). Although we show here that Cas is not required for the Chinmo/Imp to Syncrip/Broad/E93 transition, it has other functions. Cas expression in larval neuroblasts is required to establish a temporal Hedgehog gradient that ultimately triggers neuroblast cell cycle exit at pupal stages (*Chai et al., 2013*).

## How many distinct gene expression windows are present in larval neuroblasts?

Drosophila embryonic neuroblasts change gene expression rapidly, often producing just one progeny in each temporal transcription factor window (*Baumgardt et al., 2009*; *Isshiki et al., 2001*; *Moris-Sanz et al., 2014*; *Novotny et al., 2002*; *Pearson and Doe, 2003*; *Tran and Doe, 2008*). In contrast, larval neuroblasts divide ~50 times over their 120 hr lineage (*Truman and Bate, 1988*; *Yu and Lee, 2007*). Mushroom body neuroblasts make just four different neuronal classes over time (*Ito and Awasaki, 2008*; *Liu et al., 2015*; *Zhu et al., 2006*), whereas the AD (ALad1) neuroblast makes ~40 distinct projection neuron subtypes (*Kao et al., 2012*; *Lai et al., 2008*; *Yu et al., 2010*). These neuroblasts probably represent the extremes (one low diversity, suitable for producing Kenyon cells; one high diversity, suitable for generating distinct olfactory projection neurons). Here we find that larval type II neuroblasts undergo at least seven molecularly distinct temporal windows (*Figure 9*). If we assume that the graded expression of Imp (high early) and Syncrip (high late) can specify fates in a concentration-dependent manner, many more temporal windows could exist.

## An ecdysone-independent activator of syncrip?

All of the factors characterized here respond to ecdysone signaling in an all-or-none manner, with the exception of Syncrip. For example, loss of ecdysone signaling in the *ecd$^{ts}$* mutant results in persistent expression of the early factors Chinmo and Imp, and loss of expression of the late factors Broad and E93, in all central brain neuroblasts. In contrast, Syncrip is only partially reduced by loss

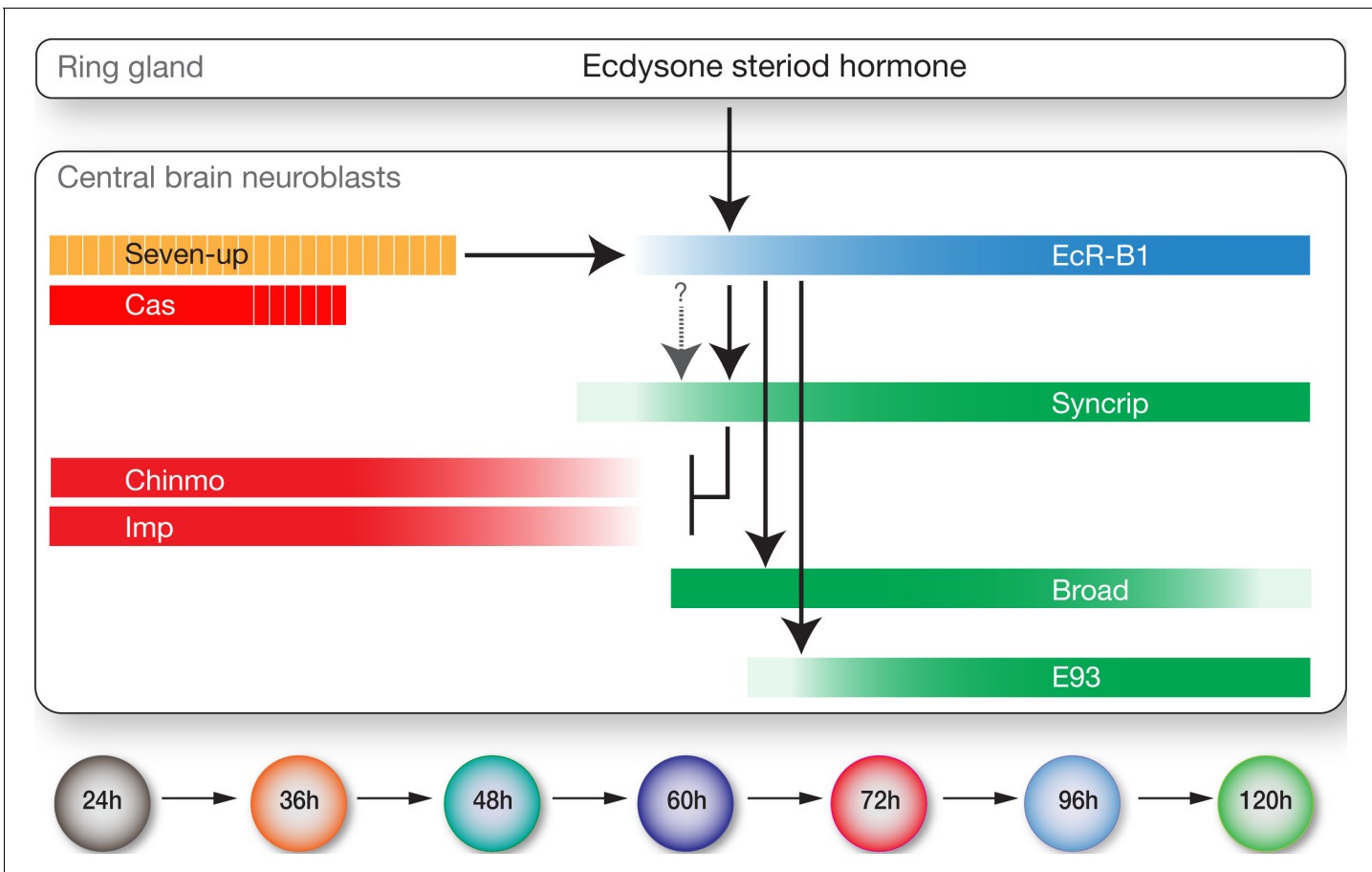

**Figure 9.** Model showing hormonal regulation of early to late temporal transitions in central brain larval neuroblasts. Summary of regulatory interactions driving larval neuroblast early-to-late temporal factor expression. Arrows indicate positive regulation; 'T' indicates negative regulation; dashed bars indicate asynchronous expression during the indicated temporal window; gradients indicate graded change in expression levels.
DOI: https://doi.org/10.7554/eLife.26287.018

of ecdysone signaling, suggesting that there is at least one additional input that drives Syncrip expression. This is supported by our finding that ~10 central brain neuroblasts express Syncrip at 36 hr and 48 hr (*Figure 8—figure supplement 1*), prior to widespread EcR-B1 neuroblast expression. *Imp* RNAi has been shown to modestly increase *syncrip* levels in the MB neuroblasts (*Liu et al., 2015*), but does not de-repress Syncrip in 24 hr type II neuroblasts (T. Lee, personal communication). We find that *Imp* mutants do not show an increase in the number of Syncrip+ type II neuroblasts at 48 hr, although the level of Syncrip protein following *imp* RNAi is elevated at 48 hr (T. Lee, personal communication). Although Imp repression of Syncrip may vary in penetrance and among different types of neuroblasts, Syncrip repression of Imp seems to be robust and conserved among all neuroblast populations tested to date (*Liu et al., 2015*; this work).

## Future directions

Our study illuminates how the major mid-larval gene expression transition from Chinmo/Imp to Broad/Syncrip/E93 is regulated; yet many new questions have been generated. What activates Svp expression in early larval neuroblasts – intrinsic or extrinsic factors? How do type II neuroblast temporal factors act together with Dichaete, Grainy head, and Eyeless INP temporal factors (*Bayraktar and Doe, 2013*) to specify neuronal identity? Do neuroblast or INP temporal factors activate the expression of a tier of 'morphogenesis transcription factors' (*Enriquez et al., 2015*) similar to leg motor neuron lineages? What are the targets of each temporal factor described here? What types of neurons (or glia) are made during each of the seven distinct temporal factor windows, and are these neurons specified by the factors present at their birth? The identification of new candidate temporal factors in central brain neuroblasts opens up the door for addressing these and other open questions.

# Materials and methods

## Fly strains

All stock numbers refer to the Bloomington stock center unless noted.

$ecd^1$ (#218) (*Garen et al., 1977*); called $ecd^{ts}$ here.
$svp^{e22}$ (*Mlodzik et al., 1990*)
$cas^{24}$ (*Cui and Doe, 1992*)
$Br^{npr3}$ FRT19A (*Kiss et al., 1976*)
$chinmo^1$ FRT40 (*Zhu et al., 2006*)
$chinmo^1$ FRT40; pointed-gal4 (this study)
$Imp^7$ (Florence Besse, CNRS, France)
$Syncrip^{f03775}$ (Exelixis collection Harvard)
Df(3L)R-G7 (# 2400)
Df(3R) BSC124 (#9289)
Df(2R) Exel6058 (#7540)
UAS-Ecr-B1$^{W650A}$ (#6872); called $EcR^{DN}$ here.
UAS-cas (Ward Odenwald, NIH)
UAS-svp (Yash Hiromi, NIG, Japan)
svp-lacZ (#26669) (*Mlodzik et al., 1990*)
cas-lacZ (also called 1532-lacZ or ming-lacZ) (*Cui and Doe, 1992*)
EcR:GFP [MI05320] (#59823)
hs-flp UAS-mcd8: GFP; FRT40, tubulin-gal80
hs-flp UAS-mcd8: GFP;; FRT82B,tubulin-gal80 (Bassem Hassan, ICM, France)
UAS-FLP actin-FRT-stop-FRT-gal4; wor-gal4 ase-gal80; UAS-mCD8: GFP (this study)
wor-gal4 ase-gal80; $svp^{e22}$, FRT82B/TM6btb (this study)
wor-gal4 ase-gal80; $cas^{24}$, FRT82B/TM6btb (this study)

## Fly genetics

To generate *svp* or *cas* mutant type II NB MARCM clones, *hs-flp UAS-mcd8: GFP;, ; FRT82B,tubulin-gal80 /TM6tb* flies were crossed to *wor-gal4 ase-gal80; Svp$^{e22}$, FRT82B/TM6, Tb* or *wor-gal4 ase-gal80; Cas$^{29}$, FRT82B/TM6, Tb* flies, respectively. *Chinmo$^1$* clones were induced by crossing *hs-flp,*

*UAS-mcd8: GFP; FRT40, tubulin-gal80* flies to *chinmo[1], FRT40; pointed-gal4* flies. *chinmo* and *castor* MARCM clones were induced during embryogenesis and analyzed at the indicated larval time point. Briefly, embryos were collected over a period of 6 hr and heat shocked in 37°C water bath for 30–40 min. After hatching, larvae were collected for 3–6 hr and reared at 25°C until the desired time point. To induce *svp* MARCM clones, 0–4 hr larvae were heat shocked in water bath for 1 hr and reared at 25°C until the desired time point.

## TU-tagging, RNA isolation, and RNA-seq

We used *wor-Gal4,ase-Gal80; UAS-UPRT/9D11* Gal80 larvae to obtain type II neuroblast and progeny expression of uracil phosphoribosyltransferase (UPRT) at 48 hr, 72 hr and 96 hr. Larvae of appropriate age were fed on food containing 0.5 mM 4-thiouracil (Sigma, St. Louis, MO) for 5 hr, dissected larval brains were pooled and stored in RNAlater (ThermoFisher, Eugene OR). RNA purification and RNA-seq was done as described previously (*Gay et al., 2014*, *Gay et al., 2013*). Briefly, larval brains were homogenized in Trizol (ThermoFisher) followed by RNA isolation and construction of cDNA libraries followed by 100 bp single end read sequencing on Illumina HiSeq 2000. For each time point three biological replicates were sequenced, which resulted in 30–40 million single end 100 bp reads for each barcoded library. Sample reads were trimmed to remove adaptor sequences using FASTX-Toolkit (Hannon lab) and then aligned to the Drosophila genome using GSNAP aligner (*Wu and Nacu, 2010*). Only uniquely aligned reads were considered for downstream differential gene expression analysis. We used HTSeq with union mode to generate gene counts from the BAM alignment files for each sample. Gene counts table were analyzed for differential gene expression by DeSeq2 method (*Love et al., 2014*). We narrowed down our candidate gene list by selecting genes which were more than two fold either enriched or depleted across the two samples, and we focused mainly on transcription factors and RNA-binding proteins having available reagents.

## Standardizing larval development at different temperatures

All larvae were grown at 25°C unless noted, and all hours after larval hatching are standardized to growth at 25°C based on published conversions: 18°C is 2.25x slower than 25°C, and 29°C is 1.03x faster than 25°C (*Powsner, 1935*).

## In vitro culturing of larval brains

In vitro cultures were set up in 24 well plates in Schneider's insect medium (Gibco) supplemented with 10% (vol/vol) heat inactivated FBS (Sigma), 0.01% (vol/vol) Insulin solution (Sigma, I0516-5ML), and 1% (vol/vol) Pen/Strep (Sigma). 1 ug/ml of 20-hydroxy-ecdysone (Sigma) was added for control conditions. Dissections were performed at 48 hr and the prothoracic glands were removed prior to culture. Brains were cultured for 24 hr and the media was changed every 12 hr.

## Immunohistochemistry

Primary antibodies were Chicken anti-GFP (1:1000, Aves), Rat anti-Dpn (1:500 Abcam, Eugene, OR), Rabbit anti-Asense (1:2000 Cheng-Yu Lee, Univ Michigan), Mouse anti-Svp (1:50 Developmental Studies Hybridoma Bank, Iowa), Guinea pig anti-Castor (1:1000 Stefan Thor, Sweden), Rabbit anti-Castor (1:1000 Ward Odenwald, NIH; subsequently distributed by Doe lab), Rat anti-Chinmo (1:500 Nick Sokol), Rabbit anti-Imp (1:500, Paul MacDonald), Mouse anti-EcR common (1:500 Carl Thummel; detects all isoforms), Mouse anti-EcR-B1 (1:2000 Carl Thummel), Mouse anti-BrC (1:100 Gregory M. Guild), Mouse anti-BrZ1 (1:100 Gregory M. Guild), Guinea pig anti-Syncrip (1:2000 Ilan Davis, UK), or Guinea pig anti-E93 (1:500 this work). Dissection and immunostaining were performed using a standard larval immune staining protocol (*Lee et al., 2006*). Larval brains were dissected in insect media (Sigma), fixed in 4% formaldehyde in PBST (PBS with 0.3% Triton X-100) for 20 min. After fixing, brains were washed in PBST for 40 min and blocked in PBST with 5% normal goat and donkey serum mix (Vector Laboratories) for 40 min. Fixed brains were transferred to the primary antibody solutions of the appropriate dilutions and incubated overnight at 4C. The next day, brains were rinsed and washed for 40 min and then incubated in secondary antibody for 2 hr at room temperature. After secondary antibody incubation, a 40 min wash was performed and brains were stored and mounted in Vectashield (Vector Laboratories).

## Confocal imaging, data acquisition, and image analysis

Fluorescent images were acquired on a Zeiss LSM 710. Larval brain cell counting was performed using the FIJI cell counter plug in, and statistical analysis (Student's T test) was done in Graph Pad Prism software. 0 hr pupal brain glia and neuron counting were by immortalizing GFP expression in type II neuroblast progeny using *UAS-FLP actin-FRT-stop-FRT-gal4; wor-gal4 ase-gal80 UAS-mCD8: GFP*, and then counting the Repo+ or Bsh+ nuclei within the GFP+ volume; nuclei were rendered using Imaris (Bitplane) and only those within the GFP+ volume were shown to eliminate expression from outside type II lineages. Figures were assembled in Illustrator (Adobe).

## Acknowledgements

We thank Sen-Lin Lai, Minoree Kohwi, Qingzhong Ren and Tzumin Lee for comments on the manuscript; Claude Desplan, Hugo Bellen, Ward Odenwald, Tzumin Lee, Carl Thummel, Stefan Thor, Nick Sokol, Paul MacDonald, Gregory Guild, Ilan Davis, Florence Besse, and Cheng-Yu Lee for reagents. We thank Qingzhong Ren and Tzumin Lee for sharing data prior to publication and comments on the manuscript. We acknowledge the Bloomington Drosophila Stock Center (NIH P40OD018537), the Vienna Drosophila RNAi Center (VDRC), TRiP at Harvard Medical School (NIH/ NIGMS R01-GM084947), and the Developmental Studies Hybridoma Bank (DSHB). This work was funded by NIH HD27056 (CQD), a Developmental Biology Training grant T32HD007348-26 (BM), and the Howard Hughes Medical Institute.

## Additional information

### Funding

| Funder | Grant reference number | Author |
| --- | --- | --- |
| National Institutes of Health | GD27056 | Mubarak Hussain Syed<br>Brandon Mark<br>Chris Q Doe |
| Howard Hughes Medical Institute | | Mubarak Hussain Syed<br>Brandon Mark<br>Chris Q Doe |
| National Institutes of Health | Developmental Biology Training grant T32HD007348-26 | Brandon Mark |

The funders had no role in study design, data collection and interpretation, or the decision to submit the work for publication.

### Author contributions

Mubarak Hussain Syed, Conceptualization, Formal analysis, Investigation, Visualization, Methodology, Writing—original draft, Writing—review and editing; Brandon Mark, Formal analysis, Investigation, Visualization, Methodology, Writing—original draft, Writing—review and editing; Chris Q Doe, Conceptualization, Formal analysis, Investigation, Methodology, Writing—original draft

### Author ORCIDs

Mubarak Hussain Syed https://orcid.org/0000-0003-2424-175X
Chris Q Doe https://orcid.org/0000-0001-5980-8029

### Decision letter and Author response

Decision letter https://doi.org/10.7554/eLife.26287.020
Author response https://doi.org/10.7554/eLife.26287.021

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
