## [Decision Letter]

Thank you for submitting your article "Hormone induction of temporal gene expression transitions in *Drosophila* brain neuroblasts" for consideration by *eLife*. Your article has been favorably evaluated by Eve Marder (Senior Editor) and three reviewers, one of whom, Hugo Bellen, is a member of our Board of Reviewing Editors. The reviewers have opted to remain anonymous.

The reviewers have discussed the reviews with one another and the Reviewing Editor has drafted this decision to help you prepare a revised submission.

This manuscript explores the temporal progression of type II neuroblasts using TU-tagging and differential RNA expression profiling to identify novel factors that could potentially influence neuronal diversity. Through these findings, novel transcription factors were found to be temporally expressed, and in addition, these transitions are mediated by an extrinsic factor, ecdysone. Although ecdysone is known to regulate many transitions including termination of neuroblast division, this is the first study to link ecdysone function with the early to late gene expression transition in neural progenitor cells. While the reviewers agree the manuscript is of interest, a number of concerns need to be addressed by the authors.

Concerns:

1) The authors do not show actual TU-tagging data referring to that it is "to be described elsewhere." It would be nice to see some of these results in the manuscript.

2) The authors describe an intriguing temporal gene cascade in type II NBs. How clear is it that all of the eight genes involved actually play a role with regards to cell fate and/or daughter/NB proliferation control? Can the authors clarify the connection between these eight genes and cell fate or proliferation control in Type II neuroblasts or provide a functional output?

3) Do the authors know what turns on Svp? If there is an easy solution to define this factor, it should be included.

4) In the Discussion, the authors state that "It would be interesting to test whether Svp expression in Type II neuroblasts can occur normally in isolated neuroblasts cultured in vitro…" Since they already show type II cell culture data in the manuscript, can they easily test this?

5) The data from the central brain (Figure 7) are very superficial and unconvincing. They should be presented better (Dpn staining) and they should be quantified (we only see ~10 neuroblasts) or the data should be removed.

6) The requirement of ecdysone/EcR in the transition of the neuroblasts is not absolute since the transition still occurs in their absence, although very delayed (120-160h). One possibility is that the ecdysone manipulations result in a delay of development, which impacts the neuroblasts as well. Alternatively, ecdysone and its receptor might not be necessary for the transition per se but are important for properly timing the transition. Moreover, when the authors use neuroblast-specific manipulations (such as the dominant negative EcR), the neuroblasts express Broad normally, although they do not express Syncrip and E93. And if one looks at Figure 7, where the authors depict "equivalent" stages of larval brains with the ecdysone temperature-sensitive mutation in the permissive and restrictive temperature, it is easy to notice that these two brains are far from being in the same age: The ecdysone manipulations might have other consequences besides delaying larval development. The authors need to address or at least to discuss this issue. One possible experiment is to generate mutant neuroblast clones for Ecr-B1 and/or Usp and address the transition of the temporal factors in a wild-type background.

7) The authors suggest that EcR temporal expression is sufficient to induce an early to late transition or that ecdysone is not the triggering factor because "it is present throughout larval development." However, ecdysone levels dramatically change during development and have multiple distinct peaks, one of which may trigger the transition. If temporal expression of EcR is responsible for the transition, this might actually be a way for all neuroblasts to be synchronized. Are Br/Syncrip expressed in a synchronous or asynchronous manner in individual central brain neuroblasts? If asynchronous, perhaps they can respond to the multiple ecdysone peaks during larval development, as long as they already express Svp?

---

## [Author Response]

*Concerns:*

*1) The authors do not show actual TU-tagging data referring to that it is "to be described elsewhere." It would be nice to see some of these results in the manuscript.*

As the reviewers request, we have added the TU-tagging results for all of the genes discussed in this manuscript (new Figure 1—figure supplement 1). Panel A is an schematic of type II neuroblast lineage; panel B shows UPRT protein expression specifically in type II neuroblasts and their progeny; and panel C shows a heat map of genes differentially expressed at least one developmental stage. Note that elevated levels of Cas and Chinmo at 72h are likely due to the large number of neuroblast progeny expressing each gene; by 72h Cas and Chinmo proteins are undetectable in type II neuroblasts (Figure 1). Importantly, these data highlighted the temporal expression of EcR, which launched the entire paper.

*2) The authors describe an intriguing temporal gene cascade in type II NBs. How clear is it that all of the eight genes involved actually play a role with regards to cell fate and/or daughter/NB proliferation control? Can the authors clarify the connection between these eight genes and cell fate or proliferation control in Type II neuroblasts or provide a functional output?*

We completely agree! However, there are only two known molecular markers for neurons or glia born at different times in the type II neuroblast lineage: Repo+ glia are made early in the lineage (Izergina et al., 2009; Bayraktar and Doe, 2013) and BsH^+^ neurons are made late in type II lineages (Bayraktar and Doe, 2013). Although analyzing all temporal genes for a role in Repo+ glial or BsH^+^ neuron specification is beyond the scope of this paper (it would require precise birthdating of these neural subtypes and lengthy clonal functional analysis), we have followed the reviewers request and added one key experiment. We express the dominant negative EcR transgene specifically in type II neuroblast lineages, along with a GFP lineage tracer, and quantify the number of early-born Repo+ glia and the number of late-born BsH^+^ neurons. Interestingly, we find reduction in EcR activity leads to loss of late-born neurons and a corresponding expansion of early-born glia (new Figure 7). This result shows that EcR is required to limit glial production and promote late-born BsH^+^ neuron production. More speculatively, this result is consistent with the Repo+ glia being born near the end of the "early" window, and that failure to switch from Imp/Chinmo to Syp/E93 leads to continued production of glia at the expense of BsH^+^ neurons.

*3) Do the authors know what turns on Svp? If there is an easy solution to define this factor, it should be included.*

This is a great question, but we have no idea what turns on Svp.

*4) In the Discussion, the authors state that "It would be interesting to test whether Svp expression in Type II neuroblasts can occur normally in isolated neuroblasts cultured in vitro…" Since they already show type II cell culture data in the manuscript, can they easily test this?*

We agree this is an interesting experiment, and we were hoping to add it to the paper. Unfortunately, we are unable to culture isolated neuroblasts from 24h larval brains without added exogenous factors such as insulin, and therefore we are unable to address the question of neuroblast-intrinsic/extrinsic cues; perhaps neuroblasts fail to come out of quiescence properly or require organism- or brain-derived survival factors. The reviewer is correct that we show in vitro culture data, but it is whole brain explants, and it is from a later stage of development (48h-60h).

*5) The data from the central brain (Figure 7) are very superficial and unconvincing. They should be presented better (Dpn staining) and they should be quantified (we only see ~10 neuroblasts) or the data should be removed.*

We have done both. We now show Dpn Ase staining to confirm type I identity, and add quantification. There is a highly significant failure to repress the early factor Chinmo and activate the late factors Broad and E93 in the absence of ecdysone signaling within type I central brain neuroblasts.

*6) The requirement of ecdysone/EcR in the transition of the neuroblasts is not absolute since the transition still occurs in their absence, although very delayed (120-160h). One possibility is that the ecdysone manipulations result in a delay of development, which impacts the neuroblasts as well. Alternatively, ecdysone and its receptor might not be necessary for the transition per se but are important for properly timing the transition. Moreover, when the authors use neuroblast-specific manipulations (such as the dominant negative EcR), the neuroblasts express Broad normally, although they do not express Syncrip and E93. And if one looks at Figure 7, where the authors depict "equivalent" stages of larval brains with the ecdysone temperature-sensitive mutation in the permissive and restrictive temperature, it is easy to notice that these two brains are far from being in the same age: The ecdysone manipulations might have other consequences besides delaying larval development. The authors need to address or at least to discuss this issue. One possible experiment is to generate mutant neuroblast clones for Ecr-B1 and/or Usp and address the transition of the temporal factors in a wild-type background.*

We appreciate these comments, and we share the reviewers’ belief that it is essential to rule out general developmental delays as the origin of the phenotypes. We have multiple experiments that address this directly, and we have added a new paragraph to the Discussion describing them as a whole: "Although a global reduction of ecdysone levels is likely to have pleiotropic effects on larval development, we have performed multiple experiments to show that the absence or delay in late temporal factor expression following reduced ecdysone signaling is not due to general developmental delay. […] Taken together, these data support our conclusion that ecdysone signaling acts directly on larval neuroblasts to promote an early-to-late gene expression transition."

*7) The authors suggest that EcR temporal expression is sufficient to induce an early to late transition or that ecdysone is not the triggering factor because "it is present throughout larval development." However, ecdysone levels dramatically change during development and have multiple distinct peaks, one of which may trigger the transition.*

Data from Kozlova and Thummel (2000) show that the ecdysone peak at 24h (L1-L2 transition) is larger than the ecdysone level at 56h. Thus, it is not just a supra-threshold response to ecdysone that activates late factor expression. Furthermore, we've shown that none of the EcR isoforms are detectable in type II neuroblasts prior to 56h (Figure 4—figure supplement 1C, D), making it very unlikely that ecdysone could act earlier than 56h when EcR-B1 is first detected in neuroblasts. We believe that it is both exposure to ecdysone and presence of the EcR in neuroblasts that is required. These two events are first coincident at 56h after larval hatching.

*If temporal expression of EcR is responsible for the transition, this might actually be a way for all neuroblasts to be synchronized. Are Br/Syncrip expressed in a synchronous or asynchronous manner in individual central brain neuroblasts? If asynchronous, perhaps they can respond to the multiple ecdysone peaks during larval development, as long as they already express Svp?*

This is a great question, and the answer is asynchronous. Individual neuroblasts express temporal factors slightly asynchronously in each brain (although in general very reproducible, e.g. E93 starts to be expressed at ~96h but never at 60h). Whether this slight asynchrony is due to spatial identity differences between neuroblasts, or due to stochastic differences in responsiveness to ecdysone remains to be determined.